# Neurodevelopmental Outcomes after Congenital Heart Disease Surgery in Infancy: A 2-Year Serial Follow-Up

**DOI:** 10.3390/children8100911

**Published:** 2021-10-13

**Authors:** Kyeong Joo Song, Min Gi Kim, Eun Jae Ko, In Young Sung

**Affiliations:** Department of Rehabilitation Medicine, Asan Medical Center, University of Ulsan College of Medicine, 88 Olympic-ro 43-gil, Songpa-gu, Seoul 05505, Korea; song135@naver.com (K.J.S.); mingisu12@naver.com (M.G.K.)

**Keywords:** congenital heart disease, neuropsychological outcome, infant, neurodevelopmental delay

## Abstract

Background: The aim of this study is to assess the neurodevelopmental status of infant patients who underwent cardiac surgery in infancy and to investigate the factors affecting the neurodevelopmental status. Methods: This retrospective study included 108 patients who underwent cardiac surgery before the age of one. We used the Bayley Scales of Infant Development II to evaluate the neurodevelopmental status. All patients were analyzed according to the presence of the syndrome. Patients without the syndrome were analyzed according to the presence of brain lesions. Results: The mean mental developmental index (MDI) and the mean psychomotor developmental index (PDI) were 76.11 ± 20.17 and 65.95 ± 18.34, respectively, in the first evaluation, and 73.98 ± 22.53 and 69.48 ± 20.86, respectively, in the second evaluation. In the subgroup analysis, no significant difference was observed between the first evaluation and the second evaluation. Conclusions: No significant difference was observed in the degree of development of the patients in the two evaluation periods. Although the presence of syndrome, brain lesion, or gestational age affected the degree of developmental delay, more than half of the patients had developmental delay in the two evaluation periods in any of the subgroup. Therefore, the necessity of early screening and early rehabilitation intervention is emphasized.

## 1. Introduction

Recently, advances in cardiac surgical techniques and intensive care of infant patients led to an increase in the long-term survival rate of patients with congenital heart disease (CHD) [1].

Despite these advances, the possibility of neurological damage is still high in children undergoing cardiac surgery [2]. If problems in reasoning, learning, executive functioning, careless and impulsive behavior, language, and social skills are caused by neurological damage, such neurodevelopment disabilities ultimately lead to difficulties with educational attainment, employability, and insurability, and may limit the quality of life as the child grows [3]. Parents of children with CHD tend to consider the development of their children to be normal despite the child having a developmental delay and may not receive early intervention services even if recommended [4]. Routine developmental screening may be necessary to determine if developmental delays are occurring and to provide early intervention.

In a previous systematic review [5], a total of 871 subjects were evaluated in 11 studies. When evaluating mental development in 11 studies of patients under three years of age using the Bayley Scales of Infant Development (second edition) (BSID-II), it was observed that the mean value was normal (the mean mental development index (MDI) was 90.3). In the motor development evaluation, the mean value was observed as mild developmental delay (the psychomotor developmental index (PDI) was reported as 78.1), indicating a normal mental development. Similar results were obtained in a later large-scale study [6] in which the neurodevelopment of 1770 children with CHD was assessed using the BSID-II.

In another study [7], with the mean Intelligence quotient (IQ) measured at the age of five, the long-term prognosis of children who underwent cardiac surgery indicated that their preschool age was lower than normal and that they had behavior and socialization difficulties. However, another study [8] on children of about five years of age reported that most of the patients had normal development. In another study [9] which looked at CHD adolescents without the genetic disorder, most of the IQ tested using the Wechsler Preschool and Primary Scale of Intelligence III (WPPSI-III) was reported to be normal. Furthermore, several studies have reported on the neurodevelopmental outcomes of children after cardiac surgery, but the long-term results have differed from study to study.

As mentioned above, various studies have focused on neurodevelopment at early childhood development or school age. However, to our knowledge, only a few studies have investigated the serial change of neurodevelopmental outcomes by aging. Therefore, the aim of this study is (1) to assess the neurodevelopmental outcomes of infant patients who have undergone congenital heart surgery during infancy; (2) to observe the changes during growth; and (3) to determine the factors that influence the development itself and the change in development.

## 2. Materials and Methods

### 2.1. Study Design and Participants

This study was approved by the Ethical Committee of Asan Medical Center (ref number: 2018-1048). This study was designed as a retrospective, single-center cohort study. The information was collected by medical chart review. Patients who underwent cardiac surgery under one year of age were screened from 1 January 2007, to December 2018. Among them, the patients who underwent two neurodevelopmental assessments using the Bayley Scales of Infant Development II (BSID-II) at 6–18 months and 19–36 months, respectively, were included. Moreover, those patients whose medical chart information was insufficient were excluded.

### 2.2. Bayley Scales of Infant Development II (BSID-II)

The BSID-II provides [10,11] a standardized assessment of the cognitive and motor development of children between 1 and 42 months of age. It has two scores: the MDI and PDI. The MDI assesses memory, problem solving, early number concepts, generalization, vocalization, language, and social skills. The PDI is a single motor scale that encompasses fine and gross motor skills. Gross motor skills refer to large movements such as crawling and running while fine motor skills refer to small movements such as picking up objects using the thumb and fingers. In this study, the Korean Bayley Scales of Infant Development (K-BSID-II) [12], modified to suit the characteristics, language, and culture of Korean infants, was used to assess infants and young children by a trained professional inspector according to a standardized procedures.

The mean ± standard deviation (SD) of the MDI and PDI in the standard population is 100 ± 15. A score of under 85 is rated below the average level of development, whereas a score of under 70 indicates a developmental delay. Furthermore, the patients were categorized into three groups according to the SD. Those with a score above 85 were classified as the normal group, those with a score of 70–84 as the mild developmental delay (DD) group, and those with a score below 69 as the significant DD group. [6,11,13] When calculating the mean index score, those children who did not achieve the minimum MDI and PDI score using the BSID-II (which is 50) were nominally assigned a score of 49.

### 2.3. Other Measurements

We collected the following information: sex, birth weight, gestational age at birth, presence of any kind of syndrome (chromosomal abnormality, microdeletion syndrome (DiGeorge syndrome), Noonan syndrome, CHARGE syndrome, Treacher Collins syndrome, Scimitar syndrome, etc.), 1-min Apgar score, 5-min Apgar score, cardiac diagnosis, age at surgery, age at neurodevelopmental assessment, number of operations, rehabilitation experience, rehabilitation duration, preoperative brain ultrasonography (US) or brain magnetic resonance imaging (MRI) finding, brain hemorrhage grade, and maternal profile (age, education). Neuronal US or MRI assessment was performed by a professional radiologist while neurodevelopment assessment (BSID-II) was performed by an experienced physical therapist and occupational therapist. Evaluation results were reviewed by a rehabilitation medicine physician. Moreover, data on the results of the Korean Wechsler Intelligence Scale for Children (fourth edition) (K-WISC-IV) was also collected.

### 2.4. Subgroups of the Patients

The patients were classified as follows. First, the patients were divided into two groups: the group with syndrome and the group without syndrome. The children without the syndromic condition were analyzed by dividing them according to the presence or absence of brain lesions. The patients without any abnormality in the brain or germinal matrix hemorrhage (GMH) (Grade I) were classified as the group without neurologic damage and with brain lesions, since GMH (Grade I) is considered to be very mild. On the other hand, the patients with GMH (Grade II~IV) were classified as the group without brain lesions [14]. When analyzing the factors associated with the neurodevelopmental outcomes, only the patients without syndrome were included.

### 2.5. Statistical Analysis

IBM SPSS Statistics 25 (SPSS Inc., Chicago, IL, USA) was used to analyze all data. A *p* value of <0.05 was considered statistically significant. The continuous variables were presented as mean ± SD, whereas the categorical variables were presented as number (%).

Normality tests were used before proceeding with the analysis. The parametric statistics were used if the variables followed a normal distribution. Nonparametric statistics were used when the variables were not normally distributed. As a repeated measurement study, the results of the first developmental evaluation and the second developmental evaluation were compared using a paired *t*-test. The chi-square test is used to analyze categorical variables, whereas the paired *t*-test is used to analyze continuous variables. The association between neurodevelopment and clinical variables was assessed via univariate logistic regression analysis. Multivariate logistic regression analysis with stepwise fashioned variable selection was performed to identify the predictors of DD. All possible predictors were included in the multivariate analysis. We additionally calculated the adjusted *p*-value with adjustment variables, which found significant differences between the two groups in baseline characteristics.

## 3. Results

### 3.1. Study Population and Baseline Characteristics

A total of 6078 patients who underwent cardiac surgery before the age of one from January 2007 to December 2018 were screened for this study. The BSID-II results of 521 patients between 6 and 18 months and those of 458 patients between 19 and 36 months were obtained. Among them, 108 patients were serially evaluated twice. The first evaluation was conducted at mean 12.54 ± 2.79 months, whereas the second evaluation was performed at mean 26.01 ± 4.21 months (Figure 1). Thirty-six patients presented with syndromes, whereas 72 patients did not. In the group with syndrome, Down syndrome was the most common (*n* = 8), followed by microdeletion syndrome (DiGeorge syndrome) (*n* = 6). Others presented with Noonan syndrome, CHARGE syndrome, Treacher Collins syndrome, Scimitar syndrome, and Edwards’ syndrome. Among the patients without syndrome, 28 patients had brain lesions based on the results of the brain US or MRI, whereas 42 patients did not have brain lesions.

The overall neurodevelopmental findings are shown in Figure 2 along with the presence of syndrome and brain lesions.

Table 1 shows the clinical characteristics of patients. Fifty-four percent of all patients were born preterm (<37 weeks’ gestation age) with a median birth weight of 1901.54 ± 1024.33 g. The group without syndrome included 50 preterm babies (69.4%), whereas the group with syndrome included nine preterm babies (25%). The proportion of preterm babies in the group without syndrome was observed to be significantly higher than that in the group with syndrome (*p* = 0.000). Mean birth weight was significantly lower in the group without syndrome (1620.36 ± 1014.037 g) than that in the group with syndrome (2463.89 ± 796.166 g) (*p* = 0.000). In the case of cardiac diagnosis, a significantly higher rate of patent ductus arteriosus (PDA) was observed in the group without syndrome (*p* = 0.008). The age of surgery was 33.24 ± 47.59 days after birth for the group without syndrome, which was earlier compared with the group with syndrome (95.83 ± 80.364 days). The mean 1-min Apgar scores of the group with syndrome and the group without syndrome were 6.45 ± 1.932 and 4.94 ± 2.536, respectively. Moreover, the mean 5-min Apgar scores of the group with syndrome and the group without syndrome were 7.95 ± 1.356 and 6.74 ± 2.055, respectively. Consequently, the 1- and 5-min Apgar scores were significantly lower in the group without syndrome (*p* = 0.016, 0.015). The number of children who underwent brain US or MRI before surgery were 30 in the group with syndrome and 70 in the group without syndrome, whereas the rates of normal findings in the group with syndrome and the group without syndrome were 63.3% and 60%, respectively. Furthermore, no significant difference was observed between the two groups (*p* = 0.811).

### 3.2. Neurodevelopment Outcomes in Children after Heart Surgery

Table 2 shows the two measurements of the BSID-II scores in all patients. For the mean MDI, no difference was observed between the two evaluations, with the first evaluation being 76.11 ± 20.17 and the second evaluation 73.98 ± 22.53 (*p* = 0.465). The mean PDI score of the first evaluation was 65.95 ± 18.34 and that of the second evaluation was 69.48 ± 20.86; no significant difference was observed between the two evaluations (*p* = 0.188).

### 3.3. Comparison of the Neurodevelopment Outcomes between the Patients with and without Syndrome

Table 3 shows the BSID-II scores of the groups with and without syndrome. The group with syndrome had significantly lower MDI and PDI scores both in the first and second evaluation compared with the group without syndrome. No significant difference was observed between the first and second evaluation of the MDI and PDI in both groups.

In the group without syndrome, the MDI of 34 (47.2%) and 37 (51.4%) patients in the first and second evaluation, respectively, indicated a normal development, whereas the PDI of 21 (29.2%) and 29 patients (40.3%) in the first and second evaluation, respectively, indicated a normal development.

We additionally calculated the adjusted *p*-value by setting the birth weight, gestational age, maternal age, cardiac diagnosis, age at surgery, the number of operations, and Apgar score as adjustment variables, which showed significant differences between the two groups in baseline characteristics. Moreover, significant differences were observed between the two groups in the first and second MDI evaluation as well as the second PDI evaluation. But in the case of the first PDI evaluation, no differences were observed when the adjustment variables were adjusted (*p* = 0.067).

### 3.4. Neurodevelopmental Outcomes in the Group of Full-Term Patients According to the Presence of the Syndrome (n = 49)

Table 4 shows the neurodevelopmental outcomes in the group of full-term patients according to the presence of the syndrome (*n* = 49). The results and the proportion between each group were observed to be almost similar to the analysis including the preterm infant.

### 3.5. Neurodevelopmental Outcomes in the Group without Syndrome According to the Presence of Brain Lesion

Table 5 shows the neurodevelopmental outcomes in the group without syndrome according to the presence of brain lesion. The patients with brain lesions had significantly lower MDI scores both in the first and second evaluation compared with the patients without brain lesions (*p* = 0.023 and 0.014, respectively). No significant difference in the PDI evaluations was observed between the groups. Moreover, no significant difference was observed between the first and second evaluation of the PDI and MDI in both groups.

As shown in Table 3, the adjusted *p*-value was calculated by setting the birth weight, gestational age, maternal age, cardiac diagnosis, age at surgery, the number of operations, and Apgar score as adjustment variables, which showed significant differences in baseline characteristics. Significant differences were observed in all evaluations between the groups according to the presence of brain lesion. No difference was observed in the PDI values between the two groups before adjustment, but a significant difference was observed between the two groups after adjustment (first evaluation *p* = 0.025 and second evaluation *p* = 0.040).

### 3.6. Factors Associated with the Neurodevelopmental Outcomes in Patients without Syndrome

In Table 6, the BSID-II values of 72 children without syndrome in the second evaluation were analyzed by dividing them into two groups based on the index score of 85: (1) normal development group and (2) delayed development group.

In the MDI of the second evaluation, 37 patients had normal development and 35 patients had DD. Based on the MDI, the preoperative US or MR diagnosis showed significant differences between the two groups in the presence of brain lesions, hemorrhage grade, and the MDI and PDI scores in the first evaluation. Intracranial hemorrhage was observed in 28 out of 30 children with brain lesion, and when analyzed according to grade, it was confirmed that the grade was lower in the normal development group. In the PDI, a significant difference was observed in birth weight and gestational age between the two groups. Birth weight was low in the delayed development group, and preterm babies accounted for a large proportion in the normal development group. Furthermore, the number of operations was significantly higher in the delayed development group. The values of the first MDI and PDI evaluation were observed to be significantly lower in the delayed development group.

## 4. Discussion

The aim of this study was to assess the neurodevelopmental outcomes of infant patients who had undergone congenital heart surgery, to observe the changes during growth, and to determine the factors affecting development. Previous studies have reported the neurodevelopmental outcomes and prognosis of children with CHD, but in many cases, only one evaluation was performed. To our knowledge, this is the first study to assess the neurodevelopmental outcomes of infant patients using two consecutive developmental assessments.

The results of the MDI and PDI evaluation of more than 50% of the patients indicated a DD. This finding is consistent with that of previous studies [15,16,17,18] involving children with and without syndrome. The mean MDI and PDI values were similar or lower compared with other studies, but this was because the analysis included children with syndrome. When comparing the results of the present study with those of other studies [19,20] on patients without syndrome, similar mean values were observed.

The prevalence of congenital heart disease in preterm infants is more than twice that in infants born at term. [21] Also, the proportion of LBW infants in children with congenital heart disease is also increasing. [22] Therefore, according to the baseline characteristics of the infants in our study, the proportion of preterm infants and low-birth-weight infants was high in the group without the syndrome. The reason for the relatively high number of term infants in the group with the syndrome is that children with the syndrome often have various complex malformations rather than the preterm status itself that are risk factors for congenital heart disease. However, even considering this, there was still a significant difference in the MDI and PDI values between the groups after adjusting for this confounder. This, in itself, did not affect the results.

In this study, based on the MDI scores, the rate of the DD of children with syndrome was significantly higher than that of children without syndrome. Although no significant difference was observed between the first and second evaluation of children without syndrome, the ratio of normal development in children without syndrome was increased (from 47.2% to 51.4% for the MDI and from 29.2% to 40.3% for the PDI). Although the follow-up period was as short as one year, this result may reflect the improvement in the child’s development as they grow. There has been no one-year follow-up study so far, but when looking at previous studies with long-term follow-up, the study results were varied, but there were studies [2,8,23] that reported normal development of preschool age children. It can be suggested that the DD was improving. However, in such studies, the presence or absence of early screening or intervention, such as rehabilitation, was not known.

When the factors of development were analyzed in children without syndrome, brain lesions had a significant effect on the MDI, whereas birth weight, gestational age (GA) at birth, and the number of operations had a significant effect on the PDI. Moreover, the presence or absence of brain lesions showed an effect on the MDI, and when the adjustment variables were adjusted, it also showed an effect on the PDI. However, even 30–40% of children without brain lesions showed delayed development in the MDI evaluation, and more than 50% showed delayed development in the PDI evaluation. These results suggest that DD cannot be excluded due to the absence of a syndrome or brain lesion. What this suggests is that DD may appear in children with overall congenital heart disease even considering several influencing factors; therefore, early examination and intervention are necessary.

Based on the results of the analyses, the mean PDI score was observed to be lower than the mean MDI score. This means that motor function is more severely affected than mental development. This finding is also consistent with that of previous studies [5,6,24]. It can be predicted that motor DD will be more severe than mental DD because of the immobilization period, such as staying in the ICU during postoperative care.

Currently, early screening or therapeutic intervention for DD is not implemented in all CHD infants who underwent cardiac surgery. Selectively, if DD is suspected, the patient is referred to a rehabilitation clinic for developmental examination.

The American Heart Association Scientific Statement [2] presents an algorithm that stratifies children with CHD at high risk for developmental disabilities and recommends neurodevelopmental assessment. In this study, it was confirmed that the CHD children had DD. Considering this aspect, it can be suggested that when encountering patients in an actual clinical environment, DD should be excluded, and early screening and intervention are required. It is thought that the intervention for rehabilitation and social and environmental support at the appropriate time will ultimately lead to the improvement of the child’s function and reduction of the social and family burden of caring for the child. In our study, in many cases, children who had DD at the first evaluation still had it at the second evaluation. (As if there was no statistically significant difference between the primary and secondary evaluations). Therefore, screening and intervention should be performed as early as possible rather than waiting for an improvement in development. Most heart surgery procedures involve sternum closure, which is stabilized after two weeks to eight weeks [25], and prone posture is also possible after that. After stabilization, at least around one year of age, a developmental test such as BSID should be performed to check if the child has DD; if so, early intervention should be carried out for the child to participate in a rehabilitation program.

The study had some limitations. First, since this is a retrospective study, selection bias may be present because it focused on children who visited the Department of Rehabilitation Medicine for postoperative developmental evaluation. Thus, a prospective study is needed. Second, it is highly likely that perioperative disease and management, such as ICU care and sepsis, had an effect on the children’s development. Therefore, further research is needed. Third, limitations in assessing the long-term outcomes were observed. However, the aim of this study was to screen the children’s neurodevelopment at an early stage and to examine its changes. We also plan to conduct a long-term study in the future.

## 5. Conclusions

Based on the findings of this study, no significant difference was observed in the degree of development in the two evaluation periods. Although considering several influencing factors, more than half of the patients had DD at two periods in all subgroups. These results suggest that DD cannot be excluded even if there is no syndrome or brain lesion; therefore, the need for early screening and early rehabilitation interventions is emphasized in all infants with congenital heart disease.

## Figures and Tables

**Figure 1 children-08-00911-f001:**
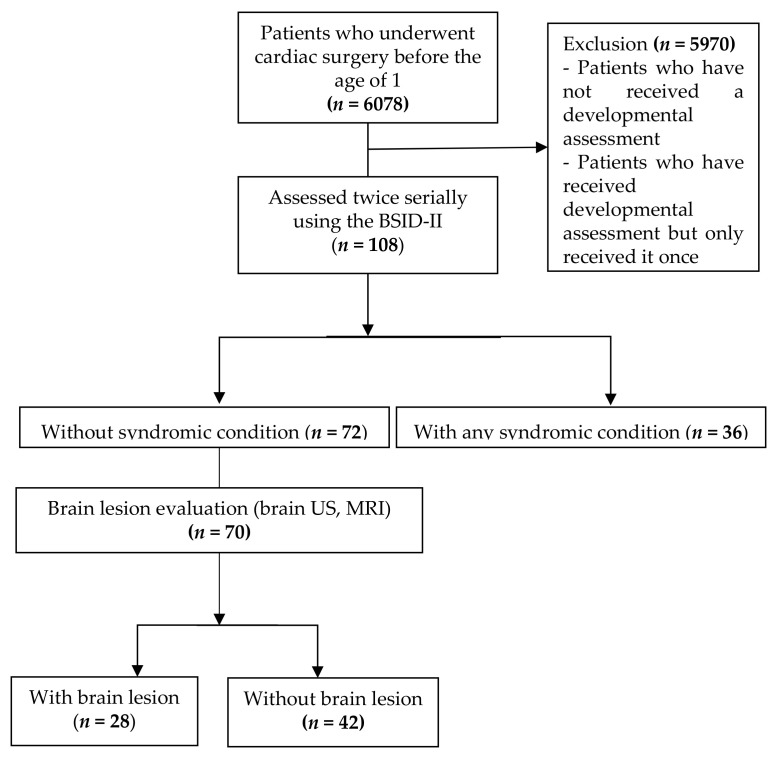
CONSORT flow diagram. BSID-II, the Bayley Scales of Infant Development (second edition); US, ultrasonography; MRI, magnetic resonance imaging.

**Figure 2 children-08-00911-f002:**
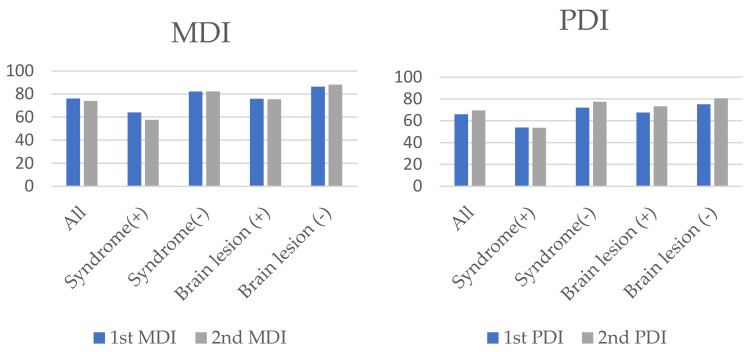
The overall neurodevelopmental findings along with the presence of syndrome and brain lesions. MDI, Mental developmental index; PDI, Psychomotor developmental index.

**Table 1 children-08-00911-t001:** Baseline characteristics.

Demographics	All Patients	Without Syndrome (*n* = 72)	With Any Syndrome (*n* = 36)	*p*-Value
Sex				
Male/female	58 (53.7)/50 (46.3)	43 (59.7)/29 (40.3)	15 (41.7)/21 (58.3)	0.077
Birth weight (g)	1901.54 ± 1024.33	1620.36 ± 1014.04	2463.89 ± 796.17	0.000 *
Normal (≥2500)	40 (36.7)	20 (27.8)	20 (55.6)	
LBW (<2500)	22 (20.2)	10 (13.9)	12 (33.3)	
VLBW (<1500)	9 (8.3)	8 (11.1)	1 (2.8)	
ELBW (<1000)	37 (33.9)	34 (47.2)	3 (8.3)	
Gestational age at birth (weeks)				0.000 *
Preterm (<37 weeks)	59 (54.1)	50 (69.4)	9 (25.0)	
Term (≥37 weeks)	49 (45.0)	22 (30.6)	27 (75.0)	
1-min Apgar score	5.25 ± 2.49	4.94 ± 2.54	6.45 ± 1.93	0.016 *
5-min Apgar score	7.01 ± 1.98	6.74 ± 2.06	7.95 ± 1.36	0.015 *
Maternal age	33.7 ± 3.90	33.19 ± 4.25	34.82 ± 2.73	0.047 *
Maternal education				
High school or higher	108 (100)	36 (100)	72 (100)	
Cardiac diagnosis				0.008 *
VSD	9 (8.3)	3 (4.2)	6 (16.7)	
PDA	42 (38.9)	37 (51.4)	5 (13.9)	
ASD	4 (3.7)	2 (2.8)	2 (5.6)	
Cyanotic	53 (49.1)	30 (41.6)	23 (63.8)	
Age at surgery (days)	53.71 ± 66.78	33.24 ± 47.59	95.83 ± 80.36	0.000 *
Age at first assessment (month)	12.49 ± 2.79	12.72 ± 2.64	12.03 ± 3.06	0.224
Age at second assessment (month)	25.80 ± 3.94	25.85 ± 4.01	25.69 ± 3.84	0.850
Number of operations	1.95 ± 1.76	1.63 ± 1.22	2.63 ± 2.41	0.005 *
Preop US or MRI (*n* = 30/70)				0.811
Nonspecific finding	61 (61)	42 (60)	19 (63.3)	
ICH, PVH	19 (19)	15 (21.4)	4 (13.3)	
Etc.	20 (20)	13 (18.6)	7 (23.4)	
Hemorrhage grade				0.124
I	18 (94.7)	14 (93.3)	4 (100)	
II	1 (5.3)	1 (6.7)	0	
III	0	0	0	
IV	0	0	0	
Rehabilitation experience (have/did not have)	98 (90.7)/10 (9.3)	66 (91.7)/6 (8.3)	32 (88.9)/4 (11.1)	0.642
Perioperative risk				
Cardiac arrest duration (*n* = 21/29)		54.52 ± 34.90	44.69 ± 36.61	0.336
ECC duration (*n* = 23/31)		114.26 ± 60.72	86.58 ± 52.43	0.079

* *p*-Value < 0.05 compare the two group. LBW, low birth weight; VLBW, very low birth weight; ELBW, extremely low birth weight; ECC, Extracorporeal circulation; Etc-Subependymal cyst, cerebellar hemorrhage. Subependymal nodule, hydrocephalus.

**Table 2 children-08-00911-t002:** BSID-II in all patients (*n* = 108).

BSID-II.	First Evaluation	Second Evaluation	*p*-Value between the Evaluation
-	N (%)	N (%)	*p*-Value
MDI			
Mean ± SD	76.11 ± 20.17	73.98 ± 22.53	0.465
Within normal limits	39 (36.1)	39 (36.1)	-
Mild DD	29 (26.9)	20 (18.5)	-
Significant DD	40 (37.1)	49 (45.4)	-
PDI			
Mean ± SD	65.95 ± 18.34	69.48 ± 20.86	0.188
Within normal limits	21 (19.4)	30 (27.8)	-
Mild DD	22 (20.4)	18 (16.7)	-
Significant DD	65 (60.1)	60 (55.5)	-

BSID-II, the Bayley Scales of Infant Development (second edition); MDI, mental developmental index; PDI, psychomotor developmental index; SD, standard deviation; DD, developmental delay.

**Table 3 children-08-00911-t003:** Comparison of the neurodevelopmental outcomes in patients according to the presence of syndrome (*n* = 108).

Group	Without SD (*n* = 72)			With Any SD (*n* = 36)			*p*-Value between Group		Adjusted *p*-Value between Group	
BSID-II	First	Second	*p*-Value	First	Second	*p-*Value	First	Second	First	Second
**MDI (*n* (%))**										
Mean ± SD	82.13 ± 19.05	82.27 ± 21.57	0.964	64.08 ± 16.88	57.68 ± 13.54	0.068	0.000 *	0.000 *	0.001 *	0.006 *
Within normal limits	34 (47.2)	37 (51.4)		5 (13.89)	2(5.56)					
Mild DD	20 (27.8)	15 (20.8)		9 (25.00)	5 (13.89)					
Significant DD	18 (25.0)	20 (27.8)		22 (61.11)	29 (80.55)					
**PDI (*n* (%))**										
Mean ± SD	72.03 ± 18.95	77.44 ± 20.40	0.101	53.80 ± 8.50	53.55 ± 9.72	0.908	0.000 *	0.000 *	0.067	0.011 *
Within normal limits	21 (29.2)	29 (40.3)		0 (0.00)	1 (2.78)					
Mild DD	18 (25.0)	16 (22.2)		4 (11.11)	2 (5.56)					
Significant DD	33 (45.8)	27 (37.5)		32 (88.88)	33 (91.66)					

* *p*-Value < 0.05 compare the evaluation and group. BSID-II, the Bayley Scales of Infant Development (second edition); MDI, mental developmental index; PDI, psychomotor developmental index; SD, standard deviation; DD, developmental delay. Adjustment variables: birth weight, gestational age, maternal age, cardiac diagnosis, age at surgery, the number of operations, and 1- and 5-min Apgar scores.

**Table 4 children-08-00911-t004:** Comparison of the neurodevelopmental outcomes in full-term patients according to the presence of the syndrome (*n* = 49).

	Without SD (*n* = 22)			With Any SD (*n* = 27)			*p*-Value between Group		Adjusted *p*-Value between Group	
BSID-II	First	Second	*p*-Value	First	Second	*p*-Value	First	Second	First	Second
**MDI (*n* (%))**										
Mean ± SD	79.32 ± 18.45	79.77 ± 20.25	0.938	62.11 ± 17.00	55.44 ± 11.80	0.100	0.005 *	0.000 *	0.006 *	0.013 *
Within normal limits	**10 (45.5)**	**10 (45.5)**		**3 (11.1)**	**1 (3.7)**					
Mild DD	6 (27.3)	5 (22.7)		5 (18.5)	2 (7.4)					
Significant DD	6 (27.3)	7 (31.8)		19 (70.4)	24 (88.9)					
**PDI (*n* (%))**										
Mean ± SD	65.82 ± 18.20	70.23 ± 16.91	0.410	53.74 ± 8.77	52.07 ± 9.38	0.503	0.022 *	0.001 *	0.000 *	0.000 *
Within normal limits	4 (18.2)	4 (18.2)		0 (0.00)	1 (3.7)					
Mild DD	6 (27.3)	8 (36.4)		4 (14.8)	1 (3.7)					
Significant DD	12 (54.5)	10 (45.5)		23 (85.2)	25 (92.6)					

* *p*-Value < 0.05 compare the evaluation and group. BSID-II, the Bayley Scales of Infant Development (second edition); MDI, mental developmental index; PDI, psychomotor developmental index; SD, standard deviation; DD, developmental delay. Adjustment variables: birth weight, gestational age, maternal age, cardiac diagnosis, age at surgery, the number of operations, and 1- and 5-min Apgar scores.

**Table 5 children-08-00911-t005:** Comparison of the neurodevelopmental outcomes in patients according to the presence of brain lesions (*n* = 72 = 42/28).

	Without Brain Lesions (*n* = 42)			With Brain Lesions (*n* = 28)			*p*-Value between Groups		Adjusted *p*-Value between Groups	
BSID-II	First	Second	*p-*Value	First	Second	*p-*Value	First	Second	First	Second
**MDI (*n* (%))**										
Mean ± SD	86.38 ± 18.01	88.07 ± 20.00	0.685	75.89 ± 19.33	75.43 ± 21.47	0.933	0.023 *	0.014 *	0.003 *	0.004 *
Within normal limits	24 (57.1)	27 (64.3)		9 (32.1)	10 (35.7)					
Mild DD	10 (23.8)	8 (19.0)		10 (35.7)	7 (25.0)					
Significant DD	8 (19.1)	7 (16.6)		9 (32.2)	11 (39.3)					
**PDI (*n* (%))**										
Mean ± SD	75.14 ± 19.84	80.43 ± 19.84	0.226	67.54 ± 17.10	73.29 ± 21.47	0.273	0.102	0.158	0.025 *	0.040 *
Within normal limits	14 (33.3)	18 (42.9)		6 (21.4)	11 (39.3)					
Mild DD	12 (28.6)	10 (23.8)		6 (21.4)	5 (17.9)					
Significant DD	16 (38.0)	14 (33.4)		16 (57.1)	12 (42.8)					

* *p*-Value < 0.05 compare the evaluation and group. BSID-II, the Bayley Scales of Infant Development (second edition); MDI, mental developmental index; PDI, psychomotor developmental index; SD, standard deviation; DD, developmental delay. Adjustment variables: birth weight, gestational age, maternal age, cardiac diagnosis, age at surgery, the number of operations, and 1- and 5-min Apgar scores.

**Table 6 children-08-00911-t006:** Factors associated with the second evaluation of neurodevelopmental outcomes in patients without syndrome.

	MDI			PDI		
Characteristics	Normal Development (*n* = 37)	Developmental Delay (*n* = 35)	*p*-Value	Normal Development (*n* = 29)	Developmental Delay (*n* = 43)	*p*-Value
Sex (male/female)	21 (56.8)/16 (43.2)	22 (62.9)/13 (37.1)	0.604	16 (55.2)/13 (44.8)	27 (62.8)/16 (37.2)	0.518
Birth weight (g)			0.889			0.019 *
Normal (≥2500)	10 (27.0)	10 (28.6)		3 (10.3)	17 (39.5)	
LBW (<2500)	5 (13.5)	5 (14.3)		4 (13.8)	6 (14.0)	
VLBW (<1500)	6 (16.2)	2 (5.7)		6 (20.7)	2 (4.7)	
ELBW (<1000)	16 (43.2)	18 (51.4)		16 (55.2)	18 (41.9)	
Gestational age (weeks)			0.511			0.011 *
Preterm (<37 weeks)	27 (73.0)	23 (65.7)		25 (86.2)	25 (58.1)	
Term (≥37 weeks)	10 (27.0)	12 (34.3)		4 (13.8)	18 (41.9)	
1-min Apgar score	5.36 ± 2.37	4.50 ± 2.67	0.157	4.52 ± 2.46	5.24 ± 2.58	0.240
5-min Apgar score	7.00 ± 1.88	6.47 ± 2.22	0.285	6.45 ± 1.90	6.95 ± 2.16	0.317
Maternal age	33.00 ± 4.17	33.40 ± 4.37	0.692			
Cardiac diagnosis			0.708			0.091
VSD	2 (5.4)	1 (2.9)		2 (6.9)	1 (2.3)	
PDA	19 (51.4)	18 (51.4)		21 (72.4)	16 (37.2)	
ASD	0 (0)	2 (5.7)		0 (0)	2 (4.7)	
Cyanotic	16 (43.2)	14 (40.0)		6 (20.7)	24 (55.8)	
Preoperative US or MRI diagnosis (*n* = 70)			0.009 *			0.597
Nonspecific findings	27 (73.0)	15 (42.9)		18 (62.1)	24 (55.8)	
Brain lesions	10 (27.0)	20 (57.1)		11 (37.9)	19 (44.2)	
Hemorrhage grade (*n* = 28)			0.004 *			0.201
I	11 (68.8)	3 (25.0)		9 (56.3)	5 (41.7)	
II	4 (25.0)	3 (25.0)		5 (31.3)	2 (16.7)	
III	0 (0)	0 (0.0)		0 (0)	0 (0)	
IIII	1 (6.3)	6 (50.0)		2 (12.5)	5 (41.7)	
Age at operation (day)	33.84 ± 45.51	32.60 ± 50.36	0.913	36.76 ± 50.54	30.86 ± 45.95	0.609
Number of operations	1.65 ± 1.42	1.60 ± 0.98	0.867	1.14 ± 0.44	1.95 ± 1.45	0.004 *

* *p*-Value < 0.05 compare the group. MDI, mental developmental index; PDI, psychomotor developmental index; SD, standard deviation; DD, developmental delay; LBW, low birth weight; VLBW, very low birth weight; ELBW, extremely low birth weight; US, ultrasonography; MRI, magnetic resonance imaging.

## Data Availability

Not applicable.

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
