# Peer review of "Neurodevelopmental Outcomes after Congenital Heart Disease Surgery in Infancy: A 2-Year Serial Follow-Up"

_children, 2021, doi:10.3390/children8100911_

Round 1
Reviewer 1 Report
Though recent advances in cardiac surgeries and intensive care after operations have dramatically improved the life prognosis of the patients with congenital heart diseases, neurodevelopmental outcomes after surgery during infancy is still under discussion. It is suggested that long duration of cardiac arrest and/or blood transfusion in cardiac surgery might be involved in the development of sequelae of brain damage. Here Song et al. have reported by the retrospective cohort study that the two series of neurodevelopmental outcomes of pediatric patients who underwent cardiac surgery before the age of 1 were not significantly different between the first and the second evaluations, and that the risk factor affecting the degree of developmental delay were the presence of syndrome, brain lesion or gestational age. Moreover, they mentioned that the more than half of the infant patients had developmental delay in any of the subgroup. Although this paper suggested an important points for prognosis of cardiac surgery during infancy, I have several concerns including study design about the manuscript as described below,
There are several issues that warrant further consideration:
Major comments
- As authors pointed out, the group without syndrome included more than half of preterm babies and mean birth weight in the group without syndrome and that the proportion of preterm babies was significantly higher in group without syndrome than that in the group with syndrome. It is considered that preterm and low birth weight are important risk factors for developmental delay. In other words, preterm and low birth weight were significant confound factors for neuronal development. If the authors would like to assess whether only the cardiac surgery before the age of 1 affect the neurodevelopment after surgery, this study should include only full-term (or near-term) patients for research subjects or should evaluate in separate groups for preterm and/or LBW babies from full-term babies for the purpose of study to assess the neurodevelopmental outcomes of infant patients with CHD who had undergone cardiac surgery.
- Why did not the issues involved in cardiac surgery, e.g. operation duration, cardiac arrest duration, blood infusion, with or without extracorporeal circulation, or stay duration in ICH, duration of ventilation etc. evaluate for neurodevelopment (as the authors mentioned in the Discussion section)? Even if this study is retrospective, these data can be obtained from the medical records.
- The authors mentioned that this is the first study using the consecutive neurodevelopmental assessments. However, no significant difference was observed between the first and the second evaluations of children. How did the authors address the importance of early screening or early intervention including rehabilitation for neurodevelopment after surgery from the results in this study? To improve the child’s function and to reduce the social and family burden of caring for the pediatric patients, please mention the proposal from the results in this paper when and how early screening and intervention be provided.
Minor comments
- Why did the authors use BSID-II instead of BSID-III?
- Who evaluated the findings of neuronal US or MRI, and assessed neurodevelopment? Were they experts in pediatric neurology? Please comment in the method section.
- Please explain all of abbreviations (e.g. GMH).
- In 4the paragraph of p.3, please collect “down syndrome” to “Down syndrome”.

Author Response
Dear Editor,
We are sincerely grateful for your thorough consideration and scrutiny of our manuscript entitled “Neurodevelopmental outcomes after congenital heart disease surgery in infancy: a 2-year serial follow-up.” Your accurate feedback has enabled us to better understand the critical issues in this paper, and we have revised the manuscript accordingly. We hope that our revised manuscript will be considered and accepted for publication in Children. We acknowledge that the scientific and clinical quality of our manuscript was improved by the scrutinizing efforts of the reviewers and editors. Point-by-point responses to the comments are provided below.
Yours Sincerely,
Dr. Ko
#1. Reviewers’ Comments & Responses
- As authors pointed out, the group without syndrome included more than half of preterm babies and mean birth weight in the group without syndrome and that the proportion of preterm babies was significantly higher in group without syndrome than that in the group with syndrome. It is considered that preterm and low birth weight are important risk factors for developmental delay. In other words, preterm and low birth weight were significant confound factors for neuronal development. If the authors would like to assess whether only the cardiac surgery before the age of 1 affect the neurodevelopment after surgery, this study should include only full-term (or near-term) patients for research subjects or should evaluate in separate groups for preterm and/or LBW babies from full-term babies for the purpose of study to assess the neurodevelopmental outcomes of infant patients with CHD who had undergone cardiac surgery.
First of all, we would like to thank the reviewer for the meticulous review.
The incidence in patients with congenital heart disease is more than twice that in preterm infants than in full-term infants. (Tanner K et al, 2005) In addition, the proportion of LBW infants among children with congenital heart disease is also increasing. (Kecskes et al, 2002) Therefore, in our study, the proportion of preterm infants was high in the group without the syndrome. The reason for the relatively high number of term infants in the group with the syndrome is that children with the syndrome often have various complex malformations rather than the preterm itself being a risk factor for congenital heart disease.
However, if only full-term infants are evaluated separately, the result would not be statistically significant because the sample size is so small. Therefore, we tried to adjust for this factor by setting it as an adjustment variable, and the adjusted p-value was presented in Table 3 as the adjusted p-value between group values.
However, as you point out since preterm and low birth weight themselves can be risk factors for a poor neurodevelopment outcome, a separate group in full-term infant analysis was added to the statistics, and this was added to the text. (Table 4)
Although we tried to analyze the full-term and normal birth weight sample, we could not analyze it because there were only three of such infants.
We have added a few sentences that have improved the Discussion section based on your comment. Thank you very much for the thoughtful review.
References:
- Tanner, K.; Sabrine, N.; Wren, C. Cardiovascular malformations among preterm infants. Pediatrics, 2005, 116(6), e833-e838. doi.org/10.1542/peds.2005-0397
- Kecskes Z; Cartwright DW. Poor outcome of very low birthweight babies with serious congenital heart disease. Arch Dis Child Fetal Neonatal Ed. 2002;87(1):F31-F33. doi:10.1136/fn.87.1.f31
- Why did not the issues involved in cardiac surgery, e.g. operation duration, cardiac arrest duration, blood infusion, with or without extracorporeal circulation, or stay duration in ICH, duration of ventilation etc. evaluate for neurodevelopment (as the authors mentioned in the Discussion section)? Even if this study is retrospective, these data can be obtained from the medical records.
Thank you for raising this important point.
Actually, the main purpose of our study was to report the overall developmental outcome and developmental change regardless of the influencing factors. Nevertheless, the reason for investigating factors such as brain lesions or syndromes is that the presence or absence of brain lesions or syndromes can often be evaluated at the prenatal period or before congenital heart disease surgery, and interventions can be considered in advance. Factors related to surgical care or critical care were not of our interest.
However, we also think that such factors may have influenced the neurodevelopment of these infants, and we already know that there are quite a few studies that focus on these factors. So, we thought it would be meaningful to include these factors in the analysis.
So, as you mentioned, we further investigated cardiac arrest duration, with or without extracorporeal circulation. Factors that were not clear on the medical chart were excluded. Cardiac arrest duration and with or without extracorporeal circulation were often not clearly recorded; so, in the case of 50 persons with cardiac arrest, 54 persons with ECC duration who have clear records, the duration was investigated and added to Table 1.
- The authors mentioned that this is the first study using the consecutive neurodevelopmental assessments. However, no significant difference was observed between the first and the second evaluations of children. How did the authors address the importance of early screening or early intervention including rehabilitation for neurodevelopment after surgery from the results in this study? To improve the child’s function and to reduce the social and family burden of caring for the pediatric patients, please mention the proposal from the results in this paper when and how early screening and intervention be provided.
Thank you for your thoughtful comment.
In our study, in many cases, children who had DD at the first evaluation still had it at the second evaluation. (As if there was no statistically significant difference between the primary and secondary evaluations). Therefore, screening and intervention should be performed as early as possible rather than waiting for an improvement in development.
In most cases, heart surgery involves sternum closure, which is stabilized after 2 weeks to 8 weeks (Clifton et al, 2020), and prone posture is also possible after that. After stabilization, at least around 1 year of age, a developmental test such as the BSID should be performed to check if the child has developmental delay, and if so, early interventions should be performed so that the child can participate in a rehabilitation program.
Having added these to the discussion, we think our manuscript has improved in quality. Thank you so much for your thoughtful comment.
References:
- Clifton A.; Cruz G.; Patel Y.; Cahalin LP.; Moore JG. Sternal precautions and prone positioning of infants following median sternotomy: A nationwide survey. Pediatr Phys Ther. 2020;32(4):339-345. doi:10.1097/PEP.0000000000000734
- Why did the authors use BSID-II instead of BSID-III?
- Who evaluated the findings of neuronal US or MRI, and assessed neurodevelopment? Were they experts in pediatric neurology? Please comment in the method section
Neuronal US or MRI assessment was performed by a professional radiologist, and the neurodevelopmental assessment (BSID-II) was performed by an experienced physical therapist and occupational therapist. Evaluation results were reviewed by a rehabilitation medicine physician. This content has been added to the Methods. Thank you for your comment.
- Please explain all of abbreviations (e.g. GMH).
Thanks for pointing this out. GMH stands for germinal matrix hemorrhage. The content has been corrected.
- In 4th paragraph of p.3, please collect “down syndrome” to “Down syndrome”.
Thanks for pointing this out. The content has been corrected.

Reviewer 2 Report
This article deals with very interesting topic because the authors retrospectively analyzed the neurodevelopmental status of infants who underwent cardiac surgery during the first year of life. In the paper, sample was divided in subgroups based on primary diagnosis (genetic syndromes and brain lesion) and they highlighted differences in developmental quotients of children with or without genetic syndromes and successively, in the sample of children without syndromes, the differences between children with or without brain damage. Although the presence of syndromes, brain lesions, or preterm birth affected the neurodevelopment, also patients without these primary diagnoses showed developmental delay. The study is well conducted, and the interpretation of the results sounds. Despite that, I have a few major and minor concerns that need to be addressed before accepting the manuscript for publication. In particular, the explication of the rationale and the description of results are improvable.
General:
According to my opinion, the most fragile point of the paper is the difficulty to highlight the innovation of the study, that is “What does the study add?”. As a matter of fact, it is widely recognized and described in literature the presence of developmental delay in children with genetic syndrome and in children with brain lesion, regardless of congenital heart disease. So, I think that authors should focus on the data of patients without syndromes and without brain lesions. Moreover, a more in-depth analysis of the development profiles about the sample without genetic syndromes and brain lesions could be useful. Also, discussion should focus primarily on this point.
Materials and Methods:
In page 2, lines 80-81, it is reported “The PDI evaluates the gross muscle function, such as crawling and walking, and fine muscle skills required in writing and the imitation of hand movements.”. I think that this is a mistake in terminology, because “gross muscle function” and “fine muscle function” are not very clear.
In page 3, line 108, it is reported the abbreviation GMH. I think that the significance is “Germinal Matrix Hemorrhage”, but it is important to explicate it.
Results:
Tables are very clear, but the text it is very laborious to follow in reading. I think that in the text could be reported only the most important results, especially about children without syndromes and without brain damage. Furthermore, the use of some graphic representation of results could help their comprehension.
Conclusion:
It would be desirable that the authors better explain what innovation in this study is and what it adds to the existing literature. Also, it is important to highlight the clinical implications of the results about early intervention for these children.
Author Response
Dear Editor,
We are sincerely grateful for your thorough consideration and scrutiny of our manuscript entitled “Neurodevelopmental outcomes after congenital heart disease surgery in infancy: a 2-year serial follow-up.” Your accurate feedback has enabled us to better understand the critical issues in this paper, and we have revised the manuscript accordingly. We hope that our revised manuscript will be considered and accepted for publication in Children. We acknowledge that the scientific and clinical quality of our manuscript was improved by the scrutinizing efforts of the reviewers and editors. Point-by-point responses to the comments are provided below.
Yours Sincerely,
Dr. Ko
#2. Reviewers’ Comments & Responses
- General
According to my opinion, the most fragile point of the paper is the difficulty to highlight the innovation of the study, that is “What does the study add?”. As a matter of fact, it is widely recognized and described in literature the presence of developmental delay in children with genetic syndrome and in children with brain lesion, regardless of congenital heart disease. So, I think that authors should focus on the data of patients without syndromes and without brain lesions. Moreover, a more in-depth analysis of the development profiles about the sample without genetic syndromes and brain lesions could be useful. Also, discussion should focus primarily on this point.
Thank you for this important comment. We agree with your comment. The initial hypothesis in this study was that the neurodevelopmental state would change. However, contrary to the hypothesis, the first and second results did not differ significantly, so the influencing factors were analyzed a little more in-depth, and the presence or absence of the syndrome and brain lesions, which are factors considered to have the greatest influence, were selected and analyzed.
So, as you said, we also think that we should focus on the data of patients without the syndrome and without brain lesions. We have added this content to the Discussion section based on your comment. Thank you very much for the thoughtful review.
- Materials and Methods: In page 2, lines 80-81, it is reported “The PDI evaluates the gross muscle function, such as crawling and walking, and fine muscle skills required in writing and the imitation of hand movements.”. I think that this is a mistake in terminology, because “gross muscle function” and “fine muscle function” are not very clear.
In page 3, line 108, it is reported the abbreviation GMH. I think that the significance is “Germinal Matrix Hemorrhage”, but it is important to explicate it.
Thank you for pointing this out.
GMH stands for germinal matrix hemorrhage. The correction has already been made.
As you said, the explanation of PDI was not very clear; so, we changed it as seen below by referring to other references.
“The PDI is a single motor scale that encompasses fine and gross motor skills. Gross motor skills refer to large movements such as crawling and running while fine motor skills refer to small movements such as picking up objects using the thumb and fingers.”
- Results: Tables are very clear, but the text it is very laborious to follow in reading. I think that in the text could be reported only the most important results, especially about children without syndromes and without brain damage. Furthermore, the use of some graphic representation of results could help their comprehension.
Thank you for your thoughtful comment.
As you said, the Tables are very clear; however, the result of enumerating their contents one by one seems not satisfactory. We corrected the important content and deleted the rest. We also tried to aid the understanding of our readers by adding a graphic representation (Figure 2). Thank you for helping us improve our content.
- Conclusion: It would be desirable that the authors better explain what innovation in this study is and what it adds to the existing literature. Also, it is important to highlight the clinical implications of the results about early intervention for these children.
Thank you for pointing this out. The conclusion was revised to emphasize early screening and early intervention, emphasizing developmental delay even in children without risk factors.

Round 2
Reviewer 1 Report
The author responded adequately to all comments except for minor comment 4 (Why did the authors use BSID-II instead of BSID-III?). As far as I know, BSID-III is the latest version.
On line 146, please replace down syndrome to Down syndrome in paragraph of 3.1. Study population and baseline characteristics.
Author Response
We are very grateful for your careful review and thoughtful suggestions regarding our manuscript. Based on the comments and suggestions, we have made the modifications on the manuscript. Below you will find our point-by-point responses to your comments:
1. Thank you for your comment. At the period of the study (from 2007 to 2018), there was no standardized Korean version of BSID-III. Korean version of BSID-III is available from 2019. Therefore BSID II was used in this study, instead of BSID-III.
2. On line 146, we replaced down syndrome to Down syndrome in paragraph of 3.1.

Reviewer 2 Report
The authors have addressed all my concerns and therefore I support publication without further changes.
Author Response
Thank you very much for your comments. It made our manuscript a lot better.